# End-to-End Autoregressive Retrieval via Bootstrapping for Smart Reply Systems

**Benjamin Towle[1], Ke Zhou[1,2]**
[1]University of Nottingham
[2]Nokia Bell Labs
{benjamin.towle, ke.zhou}@nottingham.ac.uk

## Abstract

Reply suggestion systems represent a staple component of many instant messaging and email systems. However, the requirement to produce sets of replies, rather than individual replies, makes the task poorly suited for out-of-the-box retrieval architectures, which only consider individual message-reply similarity. As a result, these system often rely on additional post-processing modules to diversify the outputs. However, these approaches are ultimately bottlenecked by the performance of the initial retriever, which in practice struggles to present a sufficiently diverse range of options to the downstream diversification module, leading to the suggestions being less relevant to the user. In this paper, we consider a novel approach that radically simplifies this pipeline through an autoregressive text-to-text retrieval model, that learns the smart reply task end-to-end from a dataset of (message, reply set) pairs obtained via bootstrapping. Empirical results show this method consistently outperforms a range of state-of-the-art baselines across three datasets, corresponding to a 5.1%-17.9% improvement in relevance, and a 0.5%-63.1% improvement in diversity compared to the best baseline approach. We make our code publicly available.[1]

## 1 Introduction

Reply suggestion, or smart reply (SR), systems are a staple component of many commercial applications such as Gmail, Skype, Outlook, Microsoft Teams, LinkedIn and Facebook Messenger. They help the user process chats and emails quicker by offering a set of canned replies which can be clicked without requiring manual typing. However, dialogue is known to be a one-to-many problem (Zhao et al., 2017; Towle and Zhou, 2022) – namely, for any given message, there are multiple possible replies. To reflect this uncertainty, systems should

present a diverse set of options to the user. For instance, given the message `How are you?`, an SR system could suggest: {`I'm good`; `Ok`; `Not great`}. Resultantly, the quality of a given reply depends not only on the message, but on the other replies in the reply set.

Several prior works explore solutions to this problem such as removing near duplicates, penalising inter-reply similarity (Deb et al., 2019), clustering by intent (Henderson et al., 2017; Weng et al., 2019), learning latent variables (Deb et al., 2019, 2021), or model-based simulation (Towle and Zhou, 2023). However, these methods share a common design choice (Figure 1A): (1) a retrieval-based Matching model, which has learned a shared embedding space between messages and replies, returns a shortlist of top scoring replies; (2) this shortlist is refined through some diversification procedure to obtain the final reply set.

Unfortunately, this assumes that the initial shortlist contains at least one good reply set. In practice, we find Matching models often search myopically, only retrieving candidates that are very similar to one another (Figure 1A). Thus, the chosen reply set often fails to reflect a diverse range of user intents, while latency constraints make more sophisticated diversification techniques or larger shortlists prohibitive (Deb et al., 2019).

An intuitive, but – to the best of our knowledge – unexplored, solution to this problem is to conduct the retrieval autoregressively, with each reply conditioned on *both* the initial message *and* the previous replies in the set. Unfortunately, this approach encounters a second problem, namely, the lack of any datasets containing (message, reply set) pairs (Towle and Zhou, 2023). In practice, SR systems are trained on individual (message, reply) pairs obtained from conversation datasets, while the task of presenting multiple diverse replies to the user is outsourced to a separate diversification module.

---

[1] https://github.com/BenjaminTowle/STAR

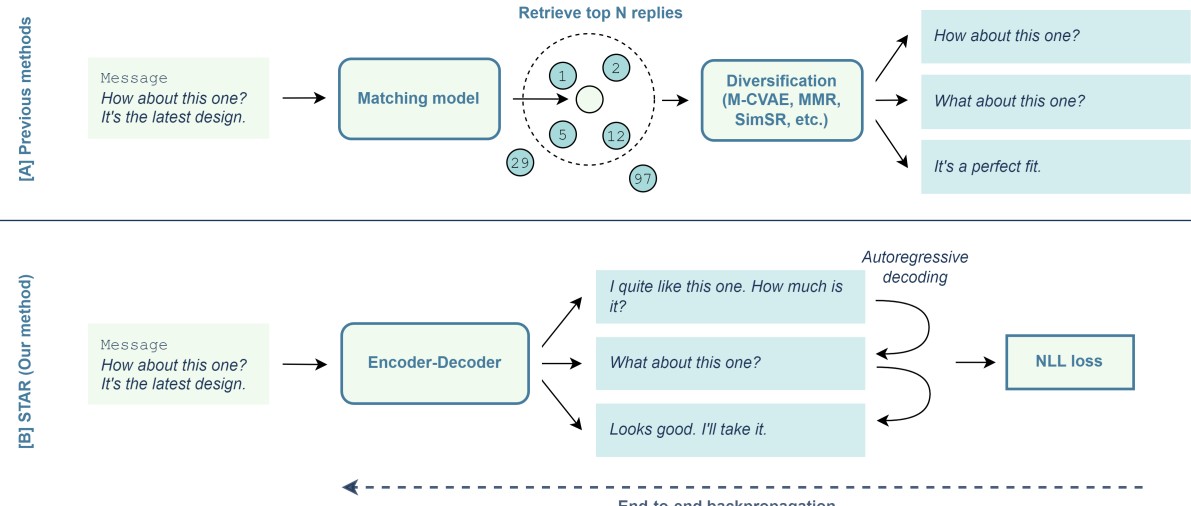

Figure 1: Previous methods [A] compared to our approach, STAR [B]. The example displayed is taken from the DailyDialog Test set, and compares the predictions of STAR with SimSR (Towle and Zhou, 2023), the next best method. Our method's suggestions present a diverse range of topics/intents to drive the conversation.

To meet this dual need, we present both (i) a bootstrapping method for creating a high-quality dataset of (message, reply sets) and (ii) a novel autoregressive retrieval model which predicts sequences of replies. For solving (i), we observe how model-based planning algorithms have been known to serve as a powerful policy improvement operator (Silver et al., 2017; Schrittwieser et al., 2019), including in several NLP systems (Jang et al., 2020, 2021). Specifically, the outputs of a model-based planning algorithm can be used to bootstrap a SR system. Further, by conducting this planning offline we are able to leverage two key advantages: (1) the system is free of the latency constraints of online inference, and therefore can increase the search space coverage of the planning algorithm; (2) the system can leverage information that would not be available during inference, such as the ground-truth reply, to further guide the search process. For (ii) we unify both steps of the standard SR pipeline into a single end-to-end model, which mitigates the myopic search, and allows the model to learn to diversify its predictions in a principled way through gradient-based learning. To this end, we present **STAR** (**S**uggested replies with **T**5 and **A**utoregressive **R**etrieval) (Figure 1B). At a high level, STAR is a text-to-text model trained to output sequences of replies, where each reply is conditioned *both* on the initial message *and* the previous replies in the sequence. Concretely, we instantiate our method with the T5 pretrained model (Raffel et al., 2020). We expand T5's vocabulary by

treating each reply in the candidate pool as a novel token, and demonstrate a simple-yet-effective technique for initialising the new token embeddings, which leverages the model's existing semantic priors. Notably, by treating each reply as a token, we limit the number of autoregressive decoding steps required, keeping the model's efficiency comparable to other retrieval-based methods.

Empirically, we evaluate our approach on three benchmarks: Reddit (Zhang et al., 2021), which is the only publicly-available SR benchmark, as well as PersonaChat (Zhang et al., 2018) and DailyDialog (Li et al., 2017) which are both widely-used in dialogue research more broadly (Zhang et al., 2019; Roller et al., 2020, *inter alia*), and share a similar conversational style with SR apps. We demonstrate superior performance over state-of-the-art baselines across all datasets, corresponding to a 5.1%-17.9% improvement in relevance, and a 0.5%-63.1% improvement in diversity compared to the best baseline approach. We further show comparable efficiency to previous methods, and perform a range of ablations to motivate our design choices.

In summary, our key contributions are as follows: (1) an autoregressive retrieval architecture for sequentially predicting suggested replies; (2) a bootstrapping framework for generating high-quality data of (message, reply set) pairs; (3) detailed analysis of model behaviour and performance including a case study and ablation of key components.

## 2 Related Work

**Smart reply**  The proprietary nature of data from email and chat applications has led several previous works to use publicly-available dialogue datasets (Zhang et al., 2021; Deb et al., 2021; Towle and Zhou, 2023) to benchmark SR methods, due to their analogous conversational nature. While early SR systems used generative models (Kannan et al., 2016), current production systems favour retrieval methods due to their greater controllability of outputs and superior latency (Deb et al., 2019). Increasing the diversity of reply suggestions is a key focus of previous work, which has been attempted by: (1) mapping replies to discrete intents / topics (Kannan et al., 2016; Chakravarthi and Pasternack, 2017; Weng et al., 2019); (2) re-weighting replies according to their similarity with other replies in the set (Carbonell and Goldstein-Stewart, 1998; Deb et al., 2019); (3) learning continuous latent variables to generate multiple queries (Zhao et al., 2017; Deb et al., 2019); (4) using model-based simulation to iteratively search and evaluate the relevance of candidate reply sets (Towle and Zhou, 2023). Our proposed method differs from all of these approaches in that our model learns to account for the interdependencies between replies through end-to-end backpropagation.

**Autoregressive retrieval**  Integrating neural retrieval into the well-established paradigm of text-to-text models is of growing interest. Earlier work focuses on outputting a document ID given a query (Tay et al., 2022). Further work has extended this by considering alternate ways of representing the document IDs, such as through unique substrings (Bevilacqua et al., 2022). Another line of work has used autoregressive retrieval for the entity linking task (Cao et al., 2021a,b,c). There, the motivation is to reduce the large number of entities by relying on the text-to-text model's pre-existing vocabulary, rather than having to retrieve embeddings from a memory-intensive dense index. Our proposed method differs considerably from these previous works both in instantiation and motivation. Instantiation-wise, we generate *multiple* replies – critical to making this possible is the novel bootstrapping technique for creating the dataset of (message, reply set) pairs to train on. Motivation-wise, our goal is to be able to condition each reply on both the input message and previous replies in the set, enabling the model to learn to predict *sequences* of replies in a differentiable way.

**Bootstrapping**  The idea of bootstrapping training data from limited resources has received significant recent interest in NLP, given the newly demonstrated few / zero-shot capabilities of many large language models (Brown et al., 2020). It has seen usage in few-shot shot text-classification (Schick and Schütze, 2021a), semantic similarity (Schick and Schütze, 2021b), tool-usage (Schick et al., 2023), retrieval (Izacard and Grave, 2021), sequence generation (He et al., 2020), and instruction-tuning (Honovich et al., 2023; Wang et al., 2023; Taori et al., 2023), amongst others. These techniques can also be seen as a form of knowledge distillation (Hinton et al., 2015), except that the training typically involves predicting the exact token targets, rather than using the soft probabilities of a teacher model. Although sometimes these techniques are used as an addition to supervised learning (He et al., 2020), in our case there are no datasets containing the ideal reply sets to suggest to the user. Instead, we must bootstrap this in a more unsupervised way, by transforming a dataset of (message, reply) pairs into a dataset of (message, reply set) pairs.

## 3 Methodology

In this section, we first describe the model-based planning process used to obtain the bootstrapped dataset of (message, reply set) pairs (Section 3.1). Then, we show how the STAR architecture can be trained on this dataset (Section 3.2).

### 3.1 Offline Dataset Creation

Our goal is to transform a dialogue dataset $\mathcal{D} = \{(x, y)\}$ of (message, reply) tuples, into a dataset $\mathcal{D}* = \{(x, Y)\}$ where $Y$ is the set of replies $\{y_k\}^K$ to be presented to the user. Algorithm 1 summarises this process. While our method is general to any arbitrary planning algorithm, we choose to instantiate our approach with a modified version of SimSR (Towle and Zhou, 2023), a recently released publicly available state-of-the-art SR method, that employs model-based simulation to predict reply sets. As the original algorithm was designed for online inference, we make several changes to benefit the offline nature of our version, and detail the full implementation below.

The initial retrieval is conducted by a Matching model $\Phi$ that separately encodes messages and replies into a shared latent space. Given an encoded

**Algorithm 1** Offline Dataset Creation. We use $N$=100, $M$=100, $\alpha$=0.75 and $\lambda$=0.05 as our default setting.

---

**Input** Matching model $\Phi$, message $x$, precomputed reply vectors $\{\mathbf{y_r}\}^R$, number of candidates $N$, number of simulations $M$, final reply set size $K$, query augmentation coefficient $\alpha$, redundancy penalty $\lambda$.

**Output** reply set $Y_K$

$\mathbf{x}, \mathbf{y} \leftarrow \Phi(x), \Phi(y)$

$\tilde{\mathbf{x}} \leftarrow \alpha\mathbf{x} + (1-\alpha)\mathbf{y}$      ▷ query augmentation

$Y_N \leftarrow N\text{-}\underset{r}{\mathrm{argmax}}(\tilde{\mathbf{x}} \cdot \mathbf{y_r})$

$Y_M \leftarrow M\text{-}\underset{r}{\mathrm{argmax}}(\tilde{\mathbf{x}} \cdot \mathbf{y_r})$

$q(y_m|x) \propto \exp \tilde{\mathbf{x}} \cdot \mathbf{y_m}$    ▷ softmax over top-M scores

$Y_G \leftarrow \emptyset$

**for** $k \leftarrow 0$ to $K$ **do**

    $y_k \leftarrow \underset{n}{\mathrm{argmax}} \sum_{m}^{M} f(Y_G^n, y_m)q(y_m|x) - \lambda f(Y_G, y_n)$

    $Y_G \leftarrow Y_G \cup y_k$

**end for**

$Y_K \leftarrow Y_G$

**return** $Y_K$

---

message $\mathbf{x} = \Phi(x)$, it retrieves the top $N$ candidates from a pool of pre-computed reply vectors $\mathbf{Y_R} = \{\mathbf{y_r}\}^R$ by combining their dot product similarity with a pre-computed language-model bias – a standard component of SR systems to downweight overly specific replies (Deb et al., 2019).

$$Y_N = N\text{-}\underset{r}{\mathrm{argmax}}(\mathbf{x} \cdot \mathbf{y_r} + \beta\mathrm{LM}(y_r)) \tag{1}$$

We then output the $K$-tuple $Y_i \in \binom{Y_N}{K}$ that has the highest expected similarity with the human reply, according to some similarity function $f(\cdot, \cdot)$.

$$\underset{i}{\mathrm{argmax}}\, \mathbb{E}_{y\sim p(\cdot|x)}\Big[f(Y_i, y)\Big] \tag{2}$$

Given the objective in SR is for at least one of the replies to be relevant, the similarity function is defined as a maximum over the sampled reply and each of the replies in the reply set, using term-level F1-score: $\underset{k}{\max} \mathrm{F1}(y_k, y)$.

We assume $y$ is sampled from the ground-truth human distribution $p(\cdot|x)$. As we do not have access to the true human distribution in practice, we instead use the same Matching model $q$ as a proxy for this, given it is trained on (message, reply) pairs. We then approximate the expectation by marginalising over the top-$M$ most likely replies:

$$\approx \underset{i}{\mathrm{argmax}} \sum_{m}^{M} f(Y_i, y_m)q(y_m|x) \tag{3}$$

In practice, it is intractable to evaluate every possible reply tuple, due to their combinatorial scaling. We therefore approximate this by greedily constructing the reply set one reply at a time. Formally,

let $Y_G$ be the set of currently selected replies, such that initially $Y_G = \emptyset$. Then, for each of $y_n \in Y_N$, we compute the expected similarity for the union of $Y_G$ and $y_n$, termed $Y_G^n = Y_G \cup y_n$ for brevity:

$$\sum_{m}^{M} f(Y_G^n, y_m)q(y_m|x) \tag{4}$$

We repeat this process for $K$ timesteps, each time appending the highest scoring reply to $Y_G$, i.e. until $|Y_G| = K$. Note that this greedy search process implicitly canonicalises the order of the replies, as selecting replies in this way causes them to be roughly ordered by individual message-reply relevance.

### 3.1.1 Adjustments

**Scaling $N$ and $M$** The original SimSR algorithm was used only in an online setting (Towle and Zhou, 2023). Therefore, the size of the search parameters $N$ (number of replies in the shortlist) and $M$ (number of simulated user replies) is kept low (15 and 25 respectively in the original paper). As we only need to run this model offline however to obtain the dataset, we find setting $N$ and $M$ to much larger values improves relevance (we use 100 for both), enabling both a broader search (i.e. by increasing $N$) and a more accurate similarity function (i.e. by increasing $M$).

**Redundancy penalty** Early testing showed that scaling the search parameters reduced diversity. We therefore introduce a redundancy penalty, which penalises the model for selecting replies that are similar to replies already in the set $Y_G$. This is analogous to the inter-document similarity penalty used in the maximum marginal relevance IR (information retrieval) technique (Carbonell and Goldstein-Stewart, 1998).

$$\sum_{m}^{M} f(Y_G^n, y_m)q(y_m|x) - \lambda f(Y_G, y_n) \tag{5}$$

**Query augmentation** Unlike during online inference, we also have access to the ground-truth reply $y$ when constructing the dataset. Previous work has found that models obtain greater representational capabilities when given access to posterior information (Paranjape et al., 2022; Towle and Zhou, 2022). We therefore use an augmented query to retrieve with the Matching model. This is obtained by interpolating between the message and ground-truth reply embeddings. This biases the model's predictions towards the observed ground-truth in

the dataset, while still allowing it to benefit from its own learned distribution.

$$\tilde{\mathbf{x}} = \alpha\Phi(x) + (1-\alpha)\Phi(y) \qquad (6)$$

## 3.2 Proposed STAR Model

We initialise STAR with a T5-based text-to-text language model, which has previously been shown to be effective in autoregressive retrieval (Tay et al., 2022). While some autoregressive retrieval approaches identify their documents/replies through unique substrings (Bevilacqua et al., 2022) or constrained beam search (Cao et al., 2021b), we focus on approaches requiring only a limited number of autoregressive steps, to maintain competitive inference speeds to existing retrieval methods (Section 5.3). There are several alternatives for this such as treating each reply set as a unique token, or separately training on each (message, reply pair), but ultimately we opted for autoregressively treating each reply as a unique token in the vocabulary in order to exploit the compositionality of reply sets (Section 5.2 for performance comparison). Note that as the types of replies used in smart reply are usually quite short and concise, e.g. 'how are you', 'I'm fine thanks', 'yes, that's right' etc., systems in deployment only need to retrieve from a pool of 30k or so replies (Deb et al., 2019), in order to provide good coverage of possible user intents. As a result, we are able to keep the size of the vocabulary reasonable. Thus, our new vocabulary is defined as: $W_{tokens} \cup W_{replies}$. An obvious challenge to this approach is that by treating each reply as a previously unseen word, it removes any semantic priors the model might have about their meaning. To mitigate this, we employ a **bag-of-words** initialisation strategy. Hence, we define the embedding of the $t$-th reply $E(y_t)$ as the average over the embeddings of the individual words within $w_n \in y_t$.

$$E(y_t) = \frac{1}{N}\sum_n^N E(w_n) \qquad (7)$$

Intuitively, this ensures that the initial embeddings are close to the word embeddings of the original vocabulary, while also capturing some of the underlying semantics of the reply. We allow the weights to update during fine-tuning. Note that for T5 the output and input embedding layers share weights, and therefore this approach is used to initialise both layers. We train the model using cross-entropy loss to predict the next reply given the current sequence of replies and messages:

$$\mathcal{L}_{NLL} = -\sum_k^K \log p(y_k|x, y_0, ..., y_{k-1}) \qquad (8)$$

## 4 Experimental Setup

### 4.1 Baselines

Previous work has largely been closed-source and is therefore unavailable for direct comparison (Henderson et al., 2017; Weng et al., 2019; Deb et al., 2019). With the exception of SimSR, which has publicly available code [2], we re-implement a variety of methods that cover the broad range of previous techniques. Due to its comparable size, all baselines apart from Seq2Seq are initialised with DistilBERT as the encoder backbone. These are summarised as follows:

**Seq2Seq** is a generative encoder-decoder. While current production systems and the majority of related works use only retrieval models (Deb et al., 2019; Towle and Zhou, 2023), at least one related work includes a standard generative transformer as a baseline (Zhang et al., 2021), which we follow here. For maximum comparability with our method, we use the same `t5-small model` as a backbone. For each message, we sample $K$ responses independently.

**Matching** represents the out-of-the-box encoder with no additional diversification strategy and was used as a baseline method by Zhang et al. (2021). It simply selects the top $K$ responses according to individual message-reply scores.

**Matching-Topic** uses an out-of-the-box topic classifier to ensure no two replies share the same topic, similar to previous work (Henderson et al., 2017; Weng et al., 2019). The classifier is trained on Twitter (Antypas et al., 2022), due to their comparable short-form open-domain chat conversations.

**Maximum Marginal Relevance (MMR)** (Carbonell and Goldstein-Stewart, 1998) is originally an IR technique, used in several previous SR works (Deb et al., 2019; Towle and Zhou, 2023), which re-weights reply scores as a linear combination of their message-reply and inter-reply similarity.

---

[2]https://github.com/BenjaminTowle/SimSR

| | Train | Valid | Test | $|\mathbf{Y_R}|$ |
|---|---|---|---|---|
| Reddit | 50k | 5k | 5k | 48k |
| PersonaChat | 66k | 8k | 8k | 64k |
| DailyDialog | 76k | 7k | 7k | 62k |

Table 1: Number of samples in the Train, Validation, Test sets and Candidate pool in the three datasets for evaluation. The Candidate pool comprises the Train set with duplicate responses removed.

**MCVAE** (Deb et al., 2019) is a conditional variational autoencoder (Zhao et al., 2017) which learns to generate multiple query vectors from a single message embedding, representing the multiple possible reply intents. Candidates are scored via a voting process, whereby the $K$ most-selected replies are chosen.

**SimSR** (Towle and Zhou, 2023) uses an iterative search and evaluation process to select possible reply sets and score them according to their expected similarity from a learned world model, which serves as a proxy for the user. To ensure comparability of SimSR with our method and the other baselines, we include the language-model bias in the scoring process (Equation 1), and also deduplicate the candidate pool.[3]

### 4.2 Datasets

We evaluate our proposed method across three datasets, summarised in Table 1. Below, we describe the datasets in more detail and motivate their inclusion. Note, other than Reddit, there are no publicly available SR datasets, due to their commercial nature (e.g. Henderson et al. (2017); Deb et al. (2019); Weng et al. (2019)). Therefore, we adopt several dialogue datasets, which is the closest alternative to conversations on proprietary chat applications.

**Reddit** (Zhang et al., 2021) was originally introduced for training multilingual SR systems, and is the only publicly available dataset specifically intended for SR purposes. As the original dataset is very large, we follow Towle and Zhou (2023) and use the reduced version of the dataset. Note, this version only contains English, as our aim is limited to the monolingual setting. Due to the organic nature of the dataset, conversations cover a very broad range of topics.

**PersonaChat** (Zhang et al., 2018) is a crowdworker-sourced dataset comprising persona-grounded conversations, in which each speaker is assigned a persona comprising a few short sentences. Following previous methods (Humeau et al., 2020), we concatenate the persona to the beginning of the message. The participants are instructed to chat naturally and to try to get to know one another.

**DailyDialog** (Li et al., 2017) is a dataset created from English language learning websites and consists of a variety of high-quality dialogues in everyday scenarios. The dataset differs from the former two in that the conversations often involve real-life scenarios, such as asking for directions, and therefore captures a different variety of conversational skills.

### 4.3 Metrics

We evaluate our method on the same weighted ROUGE ensemble as previous methods (Lin, 2004; Deb et al., 2019, 2021), which is known to correlate well with click-through rate (Zhang et al., 2021):

$$\frac{\text{ROUGE-1}}{6} + \frac{\text{ROUGE-2}}{3} + \frac{\text{ROUGE-3}}{2} \quad (9)$$

As the goal of SR systems it to ensure that at least one of the suggested replies is relevant to the user, we only record the maximum ROUGE score across each of the $K = 3$ suggested replies. We also evaluate the model on Self-ROUGE (Celikyilmaz et al., 2020): This is an unreferenced metric that measures the internal dissimilarity (i.e. diversity) within the reply set by treating one reply as the predicted reply and the other parts as the references. Note that a lower Self-ROUGE score indicates *more* diversity.

### 4.4 Inference

For inference, we use the entire training set as the candidate pool for each respective dataset, with deduplication to remove exact matches. For STAR, we greedily decode the next reply token until $K$ tokens have been decoded. Note, we only allow the model to output replies represented in the bootstrapped dataset, and also block non-replies, i.e. words from the original vocabulary, from being predicted.

## 5 Experimental Results

We focus our efforts on answering the following Research Questions: ($\mathbf{RQ_1}$) How does STAR com-

---

[3]Both changes lead to consistently improved accuracy and diversity across all datasets compared to the original paper.

pare to existing state-of-the-art methods? (Section 5.1, 5.4); (**RQ₂**) Which components of the data collection algorithm and fine-tuning have the largest impact on STAR's performance? (Section 5.2); (**RQ₃**) How efficient is STAR in inference? (Section 5.3)

## 5.1 Main Results

Table 2 compares the performance of different SR systems across the Reddit, PersonaChat and Dialy-Dialog datasets. In terms of relevance (ROUGE), STAR shows an especially large improvement in Reddit (+17.9%) and DailyDialog (+15.8%). We hypothesise the gains in PersonaChat (+5.1%) are more modest because the replies are more easily predicted due to the persona, which is concatenated to each message. This significantly reduces the noise during the initial retrieval for the baselines, as they only need to retrieve the messages relevant to that particular persona.

For diversity (Self-ROUGE), the strongest gains were found in DailyDialog (+63.1%). For PersonaChat, STAR performs much better than the retrieval methods, only falling behind Seq2Seq, due to its altogether noisier outputs as evidenced by having the worst relevance score. The Reddit results were comparatively more modest (+0.5%) – we hypothesise this is because the dataset is altogether more noisy, and so there are relatively few similar replies in the dataset, as shown by the Self-ROUGE scores being lower than the other two datasets. Overall, the consistent outperformance in both relevance and diversity metrics supports the benefits of the STAR approach.

## 5.2 Ablation

In Table 3, we conduct ablations across two key axes: data collection and STAR training. The data collection ablations serve to investigate the benefits of the novel changes to the SimSR algorithm from Section 3.1.1. The STAR training ablations investigates the degree to which the improvements in performance are caused by the bootstrapped dataset or by STAR's architecture itself; we achieve this by considering several alternative variants of STAR.

Our data collection ablations consider two features: (A) removing the query augmentation prevents the model from leveraging any ground truth information during prediction; (B) removing the redundancy penalty no longer explicitly penalises lack of diversity in predicted reply sets. For STAR training, we consider three alternative configurations:

(C) we replace the bag-of-words embeddings with randomly initialised embeddings – this removes any priors about the meaning of replies and forces the model to learn them *tabula rasa*; (D) we treat each reply set as a unique token – this removes the compositional element from the task, constraining the model to only predicting previously seen reply sets, therefore testing whether the model is capable of learning to compose novel reply sets; (E) we remove the ability to account for interdependencies between replies, by restructuring each (message, reply set) data point into $K$ data points of (message, $reply_k$), and then outputting the top-$K$ replies during inference – this investigates whether the benefit lies simply in the bootstrapped dataset being better suited to the SR task, rather than in STAR's ability to account for interdependencies between replies.

In terms of data collection ablations, we found removing the redundancy penalty significantly reduced the diversity of predictions, although in some cases offered slightly improved relevance; removing the query augmentation generally led to a worse relevance/diversity trade-off. For the variants of STAR training, we found that random embeddings consistently reduced relevance, while also led to less diverse predictions; reply sets as tokens led to the most competitive variant of STAR compared to our default setup: diversity was overall better, due to using preconstructed reply sets from the offline planning algorithm, but this came at the trade-off of reduced flexibility from being unable to construct novel reply sets when the context required it – resultantly, we saw a corresponding reduction in relevance. Finally, predicting replies separately expectedly harmed both relevance and diversity, demonstrating the importance of accounting for reply interdependencies.

In Figure 2, we further validated the individual results of our ablation by aggregating the results across datasets (applying an equal weighting to each dataset). This demonstrates the overall trend that the default STAR offers the superior trade-off between relevance and diversity, while treating reply sets as tokens offered the next best alternative. Nevertheless, we believe that keeping individual replies as tokens – thus allowing the model to construct reply sets dynamically – is likely to be an attractive property for deployed systems, enabling the overall vocabulary size to remain modest.

| Method | Reddit | | PersonaChat | | DailyDialog | |
|---|---|---|---|---|---|---|
| | ROUGE ↑ | Self-ROUGE ↓ | ROUGE ↑ | Self-ROUGE ↓ | ROUGE ↑ | Self-ROUGE ↓ |
| *Generative models* | | | | | | |
| Seq2Seq | 2.41 | 3.43 | 6.83 | **6.88*** | 4.01 | 3.91 |
| *Retrieval models* | | | | | | |
| Matching | 1.95 | 9.42 | 7.51 | 21.47 | 6.53 | 16.65 |
| M-Topic | 1.81 | 3.94 | 7.16 | 15.43 | 6.14 | 11.11 |
| M-MMR | 2.20 | 4.44 | 7.81 | 14.57 | 6.13 | 8.63 |
| M-CVAE | 2.30 | 5.02 | 7.43 | 12.21 | 6.78 | 10.49 |
| SimSR[4] | 2.79 | 2.18 | 9.04 | 10.52 | 6.82 | 4.80 |
| STAR | **3.29*** | **2.17** | **9.50*** | 7.74 | **7.90*** | **1.77*** |

Table 2: Performance of STAR across Reddit, PersonaChat and DailyDialog Test sets on relevance (ROUGE) and diversity (Self-ROUGE) metrics. **Bold** indicates best result, underline indicates second-best. * = statistically significant versus next best result on t-test with *p*-value < 0.01.

| Model Configuration | Reddit | | PersonaChat | | DailyDialog | |
|---|---|---|---|---|---|---|
| | ROUGE ↑ | Self-ROUGE ↓ | ROUGE ↑ | Self-ROUGE ↓ | ROUGE ↑ | Self-ROUGE ↓ |
| STAR | **3.35*** | 2.27 | 8.85 | 7.48 | 8.39 | 1.81 |
| *Data Collection Ablations* | | | | | | |
| A: No Query Augmentation | 2.94 | 2.00 | 8.99 | 6.94 | 7.24 | 2.89 |
| B: No Redundancy Penalty | 3.06 | 4.29 | **9.03** | 17.26 | **8.98*** | 5.90 |
| *STAR Training Variants* | | | | | | |
| C: Random embeddings | 2.67 | 4.93 | 8.39 | 10.97 | 6.84 | 4.45 |
| D: Reply sets as tokens | 2.85 | **1.59*** | 8.76 | **6.81** | 7.75 | **1.57*** |
| E: Predict replies separately | 2.20 | 26.61 | 8.07 | 30.98 | 6.43 | 20.50 |

Table 3: Performance of STAR on the Reddit, PersonaChat and DailyDialog Validation sets under different model configurations. Ablations are applied separately. **Bold** indicates best result, underline indicates second-best. * = statistically significant versus next best result on t-test with *p*-value < 0.01.

## 5.3 Run-time Efficiency

Beyond performance gains in relevance and diversity, a major advantage of an autoregressive retrieval model is the ability to leverage the scalability of GPU-based inference. Figure 3 compares the efficiency of STAR with the other baseline methods. We use an NVIDIA GeForce RTX 3060 Ti GPU and AMD Ryzen 7 5700G with Radeon Graphics CPU, with a batch size of 32. The results show that the methods can be broadly clustered into three groups. The slowest group is the generative method Seq2Seq, due to needing to generate each reply word-by-word. The middle group – SimSR, M-CVAE and M-MMR – is characterised by methods that comprise a more involved diversification pipeline. The final and fastest group includes STAR, M-Topic and Matching, where no additional post-hoc diversification is required (for M-Topic the topics can be pre-computed prior to inference).

## 5.4 Case Study

Table 4 presents a case study on the DailyDialog Test set. We compare our approach, STAR, with the top-performing baseline from Table 2, SimSR. In both examples we consistently find STAR is able to output a broader range of intents. Quantitatively, we consider the rank that each suggestion receives according to the initial retrieval of the Matching model that underlies SimSR. We see that STAR is able to perform a much more global search across the reply space, selecting replies from within the top 100 or so ranks. This would be difficult for the standard retrieve-and-rerank approach to emulate, given 100 is usually too large a number to efficiently rerank (Deb et al., 2019). Qualitatively, SimSR's suggestions converge around common phrases, e.g. 'let's go', which would be difficult to deduplicate with a heuristic rule given only a limited number of overlapping words between the replies. Conversely, STAR is able to represent a broader range of intents, such as replying with a question in both examples. Further examples are

---

[4]Results surpass reported numbers in original paper due to inclusion of language-model bias and deduplicated candidate pool to support better comparability (Section 4.1).

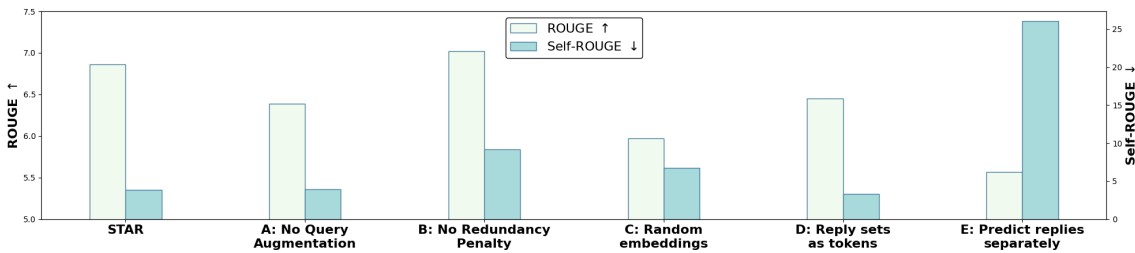

Figure 2: Comparison of overall relevance and diversity scores across ablations, obtained by averaging across all three datasets with equal weighting.

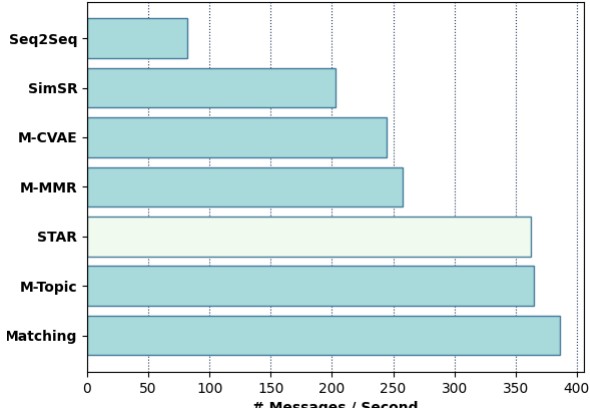

Figure 3: Comparison of run-time efficiency between STAR and the baseline methods. Results are calculated over the Reddit Validation set.

| Message: | *Hi , Kenny . Let's go for a drink .* |
|---|---|
| SimSR | - let's go ! **[#9]** 
 - ok , let's go . **[#3]** 
 - ok . let's get something to drink . **[#1]** |
| STAR | - ok . let's go . **[#5]** 
 - you want something to drink ? **[#89]** 
 - good idea . **[#105]** |
| Message: | *Of course ! Let's go .* |
| SimSR | - let's go ! **[#1]** 
 - ok , let's go . **[#5]** 
 - all right . let's go . **[#12]** |
| STAR | - let's go ! **[#1]** 
 - where are we ? **[#43]** 
 - good idea ! **[#85]** |

Table 4: Example model outputs from the DailyDialog Test set, comparing STAR (ours) with the top-performing baseline method. Numbers in **bold** indicate the ranking the reply received according to the Matching model.

provided in Appendix C.

## 6 Conclusion

We introduce **STAR**, an autoregressive retrieval system for SR, which is an end-to-end text-to-text model that sequentially predicts replies conditioned on an initial message. To train STAR, we demonstrate an approach to bootstrap a dataset of high-quality (message, reply set) pairs, from regular dialogue datasets containing only (message, reply) pairs. Empirically, our results show significant improvement over existing state-of-the-art SR baselines, across multiple datasets, corresponding to a 5.1%-17.9% improvement in relevance, and a 0.5%-63.1% improvement in diversity compared to the best baseline approach.

Future work could extend these techniques to other set-prediction tasks: e.g., in IR the relevance of each document depends on the quantity of *new* information it contains compared to other documents in the set. In recommender systems, use cases include: tailoring a user's news feed requires that the news articles presented are not simply duplicates of the same story; designing a bespoke music playlist requires songs to be unified by common themes but also sufficiently distinct from one another to maintain the listener's interest. Other lines of future work include considering alternate strategies for initialising the reply embeddings, beyond the bag-of-words initialisation demonstrated in this paper.

## Acknowledgements

We thank the reviewers for their helpful feedback and suggestions. This work is partly supported by the EPSRC DTP Studentship program. The opinions expressed in this paper are the authors', and are not necessarily shared/endorsed by their employers and/or sponsors.

## Limitations

Although our work shows that STAR is able to absorb sufficient information about the replies in its weights, this may become increasingly challenging when larger numbers of replies need to be embedded. One notable instance of this would be the multilingual setting, as many SR systems are deployed globally. In this case, each language typically has its own candidate pool. A naive implementation which creates separate reply vectors for each language would incur a significant increase in model size. In this case, we hypothesise techniques around weight-sharing between reply embeddings between languages may be beneficial, e.g. 'how are you' (en) and 'ça va' (fr) sharing the same vector. Further, our techniques are only demonstrated in publicly available datasets, whereas proprietary conversations in chat and email applications may have unique features not accounted for here (e.g. timestamps, cc and bcc information, and file attachments). Our technique also requires a planning algorithm to create the initial dataset. This theoretically creates an upper bound to the overall performance of STAR, as it is limited to cloning the behaviour of the offline planning algorithm.

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

## A  Ethical Considerations

Controlling the outputs of dialogue models is a forefront issue in ethics research for AI, particularly with the impact of recent gains in LLM capabilities. We believe the risks in the case of SR systems have several mitigants compared to this: the replies can be vetted by humans before deployment; the replies are usually in short-form, rather containing complex information the user may rely on; ultimately, a user must select one of the options, rather than the system being able to reply without user oversight. Conversely, there are some risks more unique to SR systems that should be mentioned. Particularly, the suggestions presented by the system can have subtle priming effects on user behaviour. Notably,

users have been shown to be slightly more positive in the sentiment of their emails when shown suggested replies (Wenker, 2023). SR systems are on the whole known to produce more positive sentiment messages than the human distribution (Kannan et al., 2016). We see this as an extension of the broader trend of LLMs to be overly obsequious.

## B  Implementation Details

For constructing the training dataset, we use the following hyperparameters: SimSR is initialised from the `distilbert-base-uncased` checkpoint (Sanh et al., 2019). We set the search parameters to $N = 100$ and $M = 100$. We use a redundancy penalty of $0.05$ and a blending alpha of $0.75$ for query augmentation. Both parameters provided a good trade-off between relevance and diversity in early testing, so we did not search hyperparameters further (see Section 5.2 for ablations).

For training STAR, we initialise our model with the `t5-small` checkpoint. Note that this version of T5 has a comparable parameter count to the baselines which use DistilBERT (60M versus 66M). We tokenise the dataset with a maximum message length of 64 tokens. We train our model for up to 100k steps, with a warmup of 1k steps. In practice, the model typically converged around 20k steps. We use the AdamW optimiser (Loshchilov and Hutter, 2019) with an initial learning rate of 5e-4 and linear decay. We evaluate every 2k steps, by taking the ROUGE and Self-ROUGE scores on the validation set (this uses the ground-truth from the original dataset, not from the bootstrapped dataset), and employ early stopping once both metrics have ceased to improve. Note, we found these metrics were a much more reliable stopping point than crossentropy loss, which typically converged much earlier.

## C  Further Case Studies

Table 5 displays further examples of STAR's predictions versus SimSR, taken from the PersonaChat Test set.

| | |
|---|---|
| **Persona:** | my favorite tv show is the office. |
| | i like jazz music. |
| | i do not drive because i live in new york. |
| | i am jewish. |
| | i do not eat anything sweet. |
| **Message:** | *i was but am now divorced* |
| SimSR | - i am sorry to hear that . |
| | - oh . i am sorry to hear that . |
| | - i am sorry to hear that . do you have any kids ? |
| STAR | - oh i am sorry to hear that |
| | - do you like music ? |
| | - like music ? i love jazz |
| **Persona:** | hey there my name is jordan and i am a veterinarian. |
| | love to read drama books. |
| | i love playing video games. |
| | i am also a musician on the weekends. |
| | i am originally from california but i live in florida. |
| **Message:** | *sometimes . i listen to a lot of music . do you read a lot ?* |
| SimSR | - hi how are you ? |
| | - hi how are you today |
| | - i do not like music at all |
| STAR | - i listen to music do you |
| | - no not really i like all music . |
| | - yes i do my favorite is country |

Table 5: Example model outputs from the PersonaChat Test set, comparing STAR (ours) with the top-performing baseline method. STAR is able to capture a broader range of intents through its end-to-end autoregressive retrieval.