# OpenReview forum: "End-to-End Autoregressive Retrieval via Bootstrapping for Smart Reply Systems"
_EMNLP/2023/Conference — EMNLP 2023 Findings_

### Official Review · Reviewer_86dA · 2023-07-23

**Soundness:** 3

**Excitement:**

2: Mediocre: This paper makes marginal contributions (vs non-contemporaneous work), so I would rather not see it in the conference.

**Paper Topic And Main Contributions:**

This paper presents an autoregressive retrieval method to diversify the reply set for smart reply system.
Specifically, retrieval is conducted with each reply conditioned on both the initial message and the previous replies in the set.
Several techniques are adopted to address the subsequent issues in this architecture, such as scaling, and redundancy penalty.
Experimental results on three benchmarks demonstrate improvement in relevance and diversity.

**Reasons To Accept:**

(1) The issue of chosen reply set failing to reflect a diverse range of user intents is worth investigation.

(2) Experimental results on three benchmarks demonstrate improvement in relevance and diversity.

**Reasons To Reject:**

(1) Conditioning each reply on the previous replies in the set sequentially might increase the inference latency.

(2) "Expand T5’s vocabulary by treating each reply in the candidate pool as a novel token": the augmented vocabulary might be huge due to the perplexity of sentences.

(3) bag-of-words initialisation strategy for averaging over the embeddings of the individual words might not be a good choice for representing replies.

(4) The adjustments in section 3.1.1 just combined several existing and discrete techniques for improving performance, which do not address a core research issue.

**Reproducibility:**

3: Could reproduce the results with some difficulty. The settings of parameters are underspecified or subjectively determined; the training/evaluation data are not widely available.

**Reviewer Confidence:**

3: Pretty sure, but there's a chance I missed something. Although I have a good feel for this area in general, I did not carefully check the paper's details, e.g., the math, experimental design, or novelty.

---

> ### Author Rebuttal · Authors · 2023-08-29
>
> Thanks for the helpful feedback!
> - We note that inference latency is actually lower compared to many other methods, as their handcrafted diversification techniques often impose a larger latency cost. We report these findings in Figure 2.
> - As Smart Reply systems typically cover a more limited range of intents (e.g. 'how are you', 'I'm fine thanks', 'yes, that's right' etc.) compared to fully open-domain dialogue systems, systems in deployment only need to retrieve from a pool of 30k or so replies [1], in order to provide good coverage of possible user intents. As a result, we are able to keep the size of the vocabulary reasonable. We will update the paper to include that this may be a limitation for transferring the techniques to some tasks outside of Smart Reply if they require a much larger vocabulary than this, although we believe most tasks should fall within this range.
> - Although bag-of-words embeddings outperformed random initialisation (Table 3), we recognise there may be more sophisticated initialisation techniques. We believe this is an interesting area for future work, and will update the paper to reflect this.
> - Our central motivation for combining these discrete techniques is to obtain as strong as possible a dataset in order to train our STAR model, which is one of the main contributions of this work (We note that STAR is generic and can be trained on any dataset with a <message, reply set> structure). We do this by taking advantage of the unique benefits of offline inference--lack of latency constraints and availability of ground-truth reply. Lastly, the redundancy penalty is attempting to counteract overly similar replies, which empirically (Table 3) we found to be a negative side-effect of the other two adjustments. We will update the paper to better make these motivations clear.
>
> [1] Deb et al., 2019. Diversifying reply suggestions using a matching-conditional variational autoencoder. NAACL.

---

### Official Review · Reviewer_RPm6 · 2023-08-04

**Typos Grammar Style And Presentation Improvements:** N/A
**Soundness:** 3

**Excitement:**

4: Strong: This paper deepens the understanding of some phenomenon or lowers the barriers to an existing research direction.

**Missing References:**

N/A

**Paper Topic And Main Contributions:**

This paper presents an autoregressive text-to-text retrieval model designed to learn the smart reply task from a dataset of (message, reply set) pairs obtained through bootstrapping. The experimental results demonstrate that this approach outperforms the baselines in both relevance and diversity across three datasets. The main contributions of the paper are: (1) proposing an autoregressive retrieval method for sequentially predicting suggested replies, and (2) introducing a bootstrapping framework for generating (message, reply set) pairs from (message, reply) data.

**Questions For The Authors:**

The results of SimSR presented in Table 2 differ from those in Table 3 of the original SimSR paper. Could you please clarify the differences between your re-implementations and the original paper's findings?

**Reasons To Accept:**

1. The proposed method stands out from previous approaches, which relied on matching and diversification models. Instead, the new method features an autoregressive text-to-text retrieval approach that operates in an end-to-end manner.
2. The motivation for the proposed method is transparent and the approach is both straightforward and effective. Compared to the baselines, the new method outperforms them in both relevance and diversity, while also demonstrating competitive run-time efficiency when compared to the fastest matching methods available.

**Reasons To Reject:**

1. As there is no subjective evaluation and the absolute value of ROUGE is very small, the objective evaluation alone is insufficient for accurately comparing and evaluating different models.
2. It is worth noting that all baselines were re-implemented by the authors, which may introduce some degree of unfairness when compared to previous methods.
3. The proposed method requires a SimSR model for bootstrapping the dataset, and as such, the performance may be limited by the upperbound of SimSR's performance.

**Reproducibility:**

4: Could mostly reproduce the results, but there may be some variation because of sample variance or minor variations in their interpretation of the protocol or method.

**Reviewer Confidence:**

3: Pretty sure, but there's a chance I missed something. Although I have a good feel for this area in general, I did not carefully check the paper's details, e.g., the math, experimental design, or novelty.

---

> ### Author Rebuttal · Authors · 2023-08-29
>
> Thanks for the helpful feedback!
> - We note that the results are statistically significant versus the other methods (on t-test with p-value < 0.01). The absolute value of ROUGE tends to be small in open-domain tasks such as this due to the large number of possible replies for a given message [1]. Unfortunately we do not have a deployed smart reply system to gauge click-through rate to obtain subjective evaluation, however we draw confidence from the fact that the ROUGE metrics used are reported to be well-correlated with click-through rate by previous research [1].
> - We recognise that the baselines for M-Topic, M-MMR and M-CVAE are reimplementations, due to the original methods not being publicly-available; however, for SimSR, which is the strongest-performing baseline, we use the publicly-available code. We believe the baselines selected represent the range of techniques used in Smart Reply, and there are no other publicly-available techniques that we have omitted. We will release the code of both our method and the reimplementations to enable reproducibility.
> - While in this instance the quality of our model is limited by the performance of SimSR, our STAR method is generic and can be used with any bootstrapping model. We therefore believe that discovering new bootstrapping models is an opportunity for future research. We will update the paper to reflect this.
> - We made two changes to SimSR to align it with our method for fair comparison: (i) we use the language model bias (equation 1) which is not used in original SimSR paper; (ii) we deduplicate the candidate pool (Y_R) as some basic replies, e.g. 'how are you?', occurred multiple times. These design choices closer align both methods with reported industry techniques [2]. We emphasise that these changes improved SimSR's overall performance versus the original version (e.g. ROUGE Reddit 2.40 -> 2.79, ROUGE PersonaChat 7.71 -> 9.04), and that STAR is still significantly better than the original unadjusted results from the SimSR paper. We will update the paper to make these adjustments clearer.
>
> [1] Zhang et al., 2021. A dataset and baselines for multilingual reply suggestion. ACL.
>
> [2] Deb et al., 2019. Diversifying reply suggestions using a matching-conditional variational autoencoder. NAACL.

---

### Official Review · Reviewer_YiYZ · 2023-08-05

**Soundness:** 3

**Excitement:**

3: Ambivalent: It has merits (e.g., it reports state-of-the-art results, the idea is nice), but there are key weaknesses (e.g., it describes incremental work), and it can significantly benefit from another round of revision. However, I won't object to accepting it if my co-reviewers champion it.

**Paper Topic And Main Contributions:**

To address the less relevance problem of reply suggestion from previous SR works, the author first constructed a new dataset based on the original SR dataset and trained the STAR model on the new dataset. Then, STAR based on T4 was proposed to treat each reply as a new token, and generates multiple replies in an autoregressive manner. Experimental results show that the proposed method improves both the diversity and relevance on three datasets.

**Reasons To Accept:**

* There are enough experiments to answer ``How does STAR compare to existing state-of-the-art methods?'', ``Which components of the data collection algorithm and fine-tuning have the largest impact on STAR’s performance?''  and ``How efficient is STAR in inference?''.
In addition, the author provides much discussion about the experiment results and gives reasonable explanation of several phenomena.

* The bootstrapping data construction method can get high-quality data of pairs  and improves the performance of SR task.

**Reasons To Reject:**

* The symbols in Algorithm 1 are confusing, for example,  “x tilde”  is used to represent the augmented query, but it is not used in the entire pseudocode, so how does it work?  How are Y_N and Y_M used together to select the final K replies, I have to read the paper of SimSR to figure it out.

* Why the results of SimSR in Table 2 is different from that in its original paper? Since SimSR has released its source code, what caused this difference?

* For each message in the newly created dataset, there are multiple replies. So how is the ROUGE calculated, one reference or multiple references?

* No references to Table 3 in Section 5.2. Why Reddit is chosen to perform ablation while DialyDialog is chosen for the case study?

**Reproducibility:**

4: Could mostly reproduce the results, but there may be some variation because of sample variance or minor variations in their interpretation of the protocol or method.

**Reviewer Confidence:**

3: Pretty sure, but there's a chance I missed something. Although I have a good feel for this area in general, I did not carefully check the paper's details, e.g., the math, experimental design, or novelty.

---

> ### Author Rebuttal · Authors · 2023-08-29
>
> Thanks for the helpful feedback!
> - We will adjust the methodology explanation to make it more self-contained to avoid requiring readers to be familiar with SimSR. There is a typo in the pseudocode, as x tilde should be used to obtain Y_N and Y_M instead of x.
> - We made two changes to SimSR to align it with our method for fair comparison: (i) we use the language model bias (equation 1) which is not used in original SimSR paper; (ii) we deduplicate the candidate pool (Y_R) as some basic replies, e.g. 'how are you?', occurred multiple times. These design choices closer align both methods with reported industry techniques [1]. We emphasise that these changes improved SimSR's overall performance versus the original version (e.g. ROUGE Reddit 2.40 -> 2.79, ROUGE PersonaChat 7.71 -> 9.04), and that STAR is still significantly better than the original unadjusted results from the SimSR paper. We will update the paper to make these adjustments clearer.
> - When training STAR on the newly created dataset, we use the ground-truth reply from the original dataset to calculate the ROUGE (lines 883ff.), whereby we take the maximum ROUGE between the ground-truth reply and each of the three predicted replies. We use this setup during validation as the task of predicting at least one relevant reply converged at a different rate to the task of predicting the sequence of replies from the newly created dataset. As our final evaluation setup is concerned with the former task (at least one relevant reply) we also use this to validate the model's performance during training. We will make this clearer in the paper.
> - We will update Section 5.2 to include a reference to Table 3. We find the results are consistent across all three datasets (as in the case of the main results). Due to limited space, we restricted the ablation and case study to only showing a single dataset each. We would use the additional space to expand this to all three datasets.
>
> [1] Deb et al., 2019. Diversifying reply suggestions using a matching-conditional variational autoencoder. NAACL.

---

### Meta-Review · Area_Chair_nPrY · 2023-09-23

**Recommendation:** 3

**Metareview:**

This paper presents a novel autoregressive text-to-text retrieval model designed to learn the smart reply task from a dataset using a bootstrapping method for generating data.  Extensive experiments demonstrate superior performance of the proposed method over baselines in terms of relevance and diversity.  The reviewers requested some clarifications in their reviews, which were addressed by the authors in their rebuttal.  Another concern raised was the absence of a subjective (human) evaluation, which the authors addressed to some extent by pointing to evidence of correlation between ROUGE and click-through rate.  However, the addition of a subjective human judgement evaluation of relevance and diversity might have been useful.

---

### Decision · Program_Chairs · 2023-10-07

**Decision:**

Accept-Findings

**Comment:**

This paper presents a novel autoregressive text-to-text retrieval model designed to learn the smart reply task from a dataset using a bootstrapping method for generating data.  Extensive experiments demonstrate superior performance of the proposed method over baselines in terms of relevance and diversity.  The reviewers requested some clarifications in their reviews, which were addressed by the authors in their rebuttal.  Another concern raised was the absence of a subjective (human) evaluation, which the authors addressed to some extent by pointing to evidence of correlation between ROUGE and click-through rate.  However, the addition of a subjective human judgement evaluation of relevance and diversity might have been useful.